# Psychometric properties of the Malay version of the difficulties in Emotion Regulation Scale-18 in Malaysian adolescents

**Nur Afrina Rosharudin**[1][ʘ]**, Noor Azimah Muhammad**[2][ʘ]*****, Tuti Iryani Mohd Daud**[3][‡],
**Suzana Mohd Hoesni**[1][‡]**, Siti Rashidah Yusoff**[1][‡]**, Mohamad Omar Ihsan Razman**[1][‡],
**Manisah Mohd Ali**[4][‡]**, Khairul Farhah Khairuddin**[4][‡]**, Dharatun Nissa Puad Mohd Kari**[5][‡]

**1** Centre for Research in Psychology and Human Well-Being, Universiti Kebangsaan Malaysia, Bangi, Selangor, Malaysia, **2** Department of Family Medicine, Faculty of Medicine, Universiti Kebangsaan Malaysia, Cheras, Kuala Lumpur, Malaysia, **3** Department of Psychiatry, Faculty of Medicine, Universiti Kebangsaan Malaysia, Cheras, Kuala Lumpur, Malaysia, **4** Centre for Research in Education and Community Wellbeing, Faculty of Education, Universiti Kebangsaan Malaysia, Bangi, Selangor, Malaysia, **5** Department of Counselor Education and Counseling Psychology, Faculty of Educational Studies, Universiti Putra Malaysia, Serdang, Selangor, Malaysia

ʘ These authors contributed equally to this work.
‡ TIMD, SMH, SRY, MOIR, MMA, KFK and DNPMK also contributed equally to this work.
* drazimah@ppukm.ukm.edu.my

## Abstract

### Objective

The Difficulties in Emotion Regulation Scale-18 (DERS-18) is an instrument used to measure deficits in emotion regulation. However, the instrument has not been adapted to Malaysians and has never been validated in the Malay language. This study aimed to examine the psychometric properties of the Malay version of DERS-18.

### Method

The DERS-18 underwent forward-backward translation and assessment of face and content validity. Both Malay version of the DERS-18 and DASS-21 were completed by 701 adolescents (44.4% boys) aged 13 and 14 years old. To assess its dependability, a floor and ceiling effect evaluation and Cronbach's analysis were both performed. A series of confirmatory factor analyses (CFA), bivariate correlation, and regression were performed to evaluate the construct and criterion validity, respectively.

### Results

The Malay version of DERS-18, after excluding "Awareness", indicated excellent reliability (Cronbach's α = 0.93), and acceptable internal consistency for each subscale (range of α from 0.63 to 0.82). Floor or ceiling effects were observed at item level and subscale level, but not at total level. CFA results revealed that the Malay version of the DERS-18 bifactor model (excluding "Awareness") portrayed the best construct validity ($\chi^2/df$ = 2.673, RMSEA = 0.049, CFI = 0.977, TLI = 0.968) compared to a single factor, a correlated factor, and a higher-order factor model. The DERS-18 subscales (except "Awareness") and DERS-18

**Data Availability Statement:** Regarding the Data Availability Statement, we understand the requirement for transparent data sharing. While we

have provided a minimal anonymized dataset in the Supporting information, we acknowledge that the complete dataset should be made accessible to other researchers. To ensure compliance with the PLOS One Data Availability policy, we propose the following: A) The minimal anonymized dataset will remain publicly available in the Supporting information. B) The complete dataset, including potentially sensitive information, will be made accessible upon request. C) To facilitate data access, we had a data access committee consisting of the corresponding author, Noor Azimah Muhammad (drazimah@ppukm.edu.my), the main author, Nur Afrina Rosharudin (p111600@siswa.ukm.edu.my), one of the authors, Suzana Mohd Hoesni (smh@edu.ukm.my) and a designated group representative, TRGS Let's Get R.E.A.L (trgsletsgetreal@gmail.com).

**Funding:** Manisah Mohd Ali received a fund from the Ministry of Higher education (MoHE) Malaysia (Project code: TRGS/1/2020/UKM/01/4/1). The funder had no role in study design, data collection and analysis, decision to publish, or preparation of the manuscript.

**Competing interests:** The authors have declared that no competing interests exist.

total scores were significantly correlated with stress, anxiety, and depression in a positive direction ($r$ ranged from 0.62 to 0.64, $p < 0.01$). The general factor of the DERS-18 and its specific factors ("Clarity", "Goals", and "Non-Acceptance") significantly predicted the symptoms of stress, anxiety, and depression ($R^2$ ranged from 0.44 to 0.46, $p < 0.001$).

## Conclusion

The Malay version of the DERS-18, excluding "Awareness", possessed good reliability, construct validity, and criterion validity to assess emotion dysregulation among Malaysian adolescents.

## Introduction

According to Gross [1], emotion regulation is the process of influencing emotions to control when humans experience them and how they are conveyed. Additionally, emotion regulation refers to an individual's capacity to identify, analyse, and respond to their own emotions [2]. Alternatively, Gratz and Roemer [3] suggested that emotion regulation involves (a) understanding one's emotions, (b) accepting one's emotions, (c) controlling impulsive behaviours by behaving in accordance with one's desired goals, and (d) using adaptive strategies of emotion regulation. These processes are necessary for regulating emotion successfully [4], and their misuse or deficits would indicate difficulties in emotion regulation, or emotion dysregulation [3]. Emotion dysregulation has become a major area of research that attempts to explain the onset and maintenance of psychopathologies such as eating disorders [5], emotion disorders [6] and depression [7, 8].

One questionnaire that is commonly used to assess emotion dysregulation is the Difficulties in Emotion Regulation Scale (DERS) by Gratz and Roemer [3]. The DERS was created on the basis of prominent clinical conceptualisations of emotion regulation and aimed to identify the processes of emotion regulation that do not function well [9]. It comprises 36 items with six subscales, namely the lack of awareness in one's emotional states (Awareness), nonacceptance of emotion response (Acceptance), the lack of clarity in emotion response (Clarity), difficulties in engaging goal-directed behaviours (Goals), difficulties in controlling impulse (Impulse), and limited access to effective emotion regulation strategies (Strategies). The DERS-36 has been translated into multiple languages [10–12] and validated in non-clinical [3, 10–15] and clinical samples [16, 17].

A shorter version of the DERS, called the DERS-18 was developed by retaining 18 out of its original 36 items [4]. Victor and Klonsky [4] recruited American samples from diverse contexts (i.e., adolescents, inpatient, young adults, and adults) to examine the psychometric properties of DERS-18. Three items with the highest factor loadings on each subscale of DERS-36 were chosen based on the exploratory factor analysis (EFA) employed in the study. Despite a reduction in items, the six subscales from the original DERS, namely "Awareness", "Acceptance", "Clarity", "Goals", "Impulse", and "Strategies" were retained. In other studies, it has been replicated across age groups including adolescents and adults [18–20]. In addition, it has also been validated in various samples, including Chinese [21], minority [22], and American samples [18–20, 23], and has demonstrated a high reliability of between 0.87 and 0.97 [4, 18, 19, 22–24]. Furthermore, it demonstrated acceptable to good internal consistency for "Awareness" (ranging from 0.58 to 0.92), "Clarity" (ranging from 0.53 to 0.92), "Non-Acceptance" (ranging from 0.80 to 0.96), "Goals" (ranging from 0.83 to 0.95), "Impulse" (ranging from 0.86 to 0.93), and "Strategy" (ranging from 0.75 to 0.94) [4, 18, 19, 22–24].

The factor structure of DERS is one of the significant concerns. Although the original study of DERS [3] and other studies proposed a six-factor structure [18, 21, 25], the utility of the Awareness subscale was inconsistently demonstrated in subsequent studies. For example, Semborski et al. [20] reported that a correlated model without "Awareness" fit their homeless sample adequately. Similarly, Lee et al. [14] and Medrano and Trogol [26] examined a non-clinical sample and found the five-factor correlated model to be an acceptable fit. Using a clinical sample, a study that utilised a bifactor model found a five-factor model without "Awareness" also fit their data [23]. Although Mekawi et al. [22] discovered that the original six-correlated factor model suited a sample of black women adequately, their findings indicated that the five-correlated models without "Awareness" fit their data the best. Overall, these contradictory findings imply that future research is required to investigate the applicability of "Awareness" in conceptualising emotion dysregulation [13, 17].

The dimensionality of DERS, whether unidimensional (use of total scores) or multidimensional (use of subscale scores), is another area of concern. In numerous studies, the unidimensional model of DERS exhibited poor fit [10, 13, 21, 22, 27]. Several investigations evaluated the multidimensional DERS test found that the six-correlated model fit their data well [13, 14, 21, 22]. Despite the findings, several studies used a higher-order model and a bifactor model to justify reporting the DERS as an aggregate scale. For example, in the studies of Zhao et al. [21] and Mekawi et al. [22], a higher-order model fit their data adequately. Bardeen et al. [13] also discovered that a higher-order model provided an adequate fit. However, in Lee et al.'s study [14], a higher-order model did not fit well.

Various studies examined the adequacy of the bifactor model have yielded mixed results. Mekawi et al. [22] suggested that the correlated traits model demonstrated superior fit when compared to the bifactor model. Meanwhile, Zhao et al. [21] and Hallion et al. [23] reported that the bifactor model was a good fit for their respective datasets. In McVey's [27] study, a bifactor model that exhibits a correlation between "Awareness" and "Clarity" provided the best fit compared to other models. Generally, the assessment of the DERS dimensionality is crucial due to the prevalence of using both total scores and scale scores in research as evidenced by Darmadi and Badayai [28], and Murad, Kamaluddin and Mohd Nasir [29]. This practice is both useful and frequently employed [21].

To increase understanding of emotion dysregulation among adolescents, a tool that is applicable and suitable for the Malaysian population is required. Even though emotion regulation studies have been conducted in Malaysia, they have focused on strategies to regulate emotion [30–35]. Nevertheless, an assessment instrument for emotion regulation difficulties has not been validated. Deficits in emotion regulation, a prevalent predictor of psychopathological symptoms [7, 8], should be the subject of additional research in Malaysia. Therefore, the current study sought to examine the Malay-translated DERS-18 and its psychometric properties in adolescent samples. First, an evaluation of floor and ceiling effects was conducted. Then, the internal consistency and reliability of DERS-18 and its subscales were examined. The factor structure and dimensionality of DERS-18 were then examined, as tested in the previous studies [10, 13, 14, 17, 18, 20–23, 27]. Finally, the criterion validity of DERS 18 was tested with psychopathological symptoms of depression, anxiety, and stress.

## Method

### Participants

The original sample comprised 701 (311 boys and 389 girls) adolescents from ten public schools in Malaysia. The participants' ages ranged from 13 to 14 years (M = 13.5, SD = 0.5). This stage of early adolescence is known to have a higher variability of negative emotions than

late and middle adolescence [36, 37]. This emotional variation is related to the biological changes, hormonal changes and reward sensitivity that they experience [38].

A proportionate random stratified sampling technique was used to recruit the adolescents. This technique was chosen to overcome the diversity of the adolescent population, which comprises different races and religions [39]. The population was first divided into four strata based on location, namely the north, south, east, and west regions of Peninsular Malaysia. In accordance with the Department of Statistics Malaysia [39], this study planned to recruit 25.1% of its sample size from the north region, 23.2% from the south region, 17.0% from the east region, and 33.7% from the west region. Then, one or two public schools would be randomly chosen from each stratum, and the representative teacher from each school would randomly select students who met the inclusion criteria.

Inclusion criteria include students with strong Malay reading, writing and comprehension skills who have received parental consent. Adolescents with special needs, cognitive impairments, illiterate, and those under psychological treatment were excluded. In the results section, the final sample size (after excluding missing data) and the demographic characteristics of participants were summarised.

## Instruments

*The Malay version of the Difficulties in Emotion Regulation Scale-18 (DERS-18).* The initial instrument used to measure difficulties in emotion regulation was The Difficulties in Emotion Regulation Scale (DERS) by Gratz and Roemer [3]. It is a self-reported questionnaire consisting of six subscales: awareness, clarity, impulse, goals, non-acceptance, and strategies. The "Awareness" subscale aims to measure the deficiency in awareness of one's emotion; the "Clarity" subscale is to measure the deficiency in clarity about one's emotion; the "Impulse" subscale is to measure the deficiency in ability to manage one's impulses during negative emotion; the "Goals" subscale is to measure the deficiency in ability to engage in goal-directed behaviours during negative emotions; the "Non-Acceptance" subscale is to measure the deficiency in accepting of one's emotion; and lastly, the "Strategies" subscale is to measure the deficiency in accessing effective strategies during negative emotions.

Each subscale originally contained six items [3], which Victor and Klonsky [4] subsequently reduced to 18 items. Each subscale of DERS-18 consists of three items from the DERS-36. A five-point Likert scale is employed to evaluate responses from 1 (almost never) to 5 (almost always) in the Malay version of DERS-18. Using the same criteria as the original developers [3, 4], all awareness items scores were reversed. Total scores and total subscale scores are determined by adding the scores of the corresponding items, and higher scores imply emotional regulation difficulties.

*The Malay version of the Depression, Anxiety and Stress Scale-21 (DASS-21).* Musa, Fadzil, and Zain [40] have validated The Malay version of DASS-21. This open-access survey consists of 21 items that measure stress, anxiety, and depression, and these items are related to the participants' recent emotional states. Seven items are rated on a 4-point Likert scale ranging from 0 (did not apply to me) to 3 (applied to me very much) for each subscale. Higher total subscale scores indicate increased symptomatology. Previous research has demonstrated that the reliability and validity of the Malay version of the DASS-21 are excellent. Cronbach's values (α) of 0.95, 0.85, and 0.87, respectively were found for depression, anxiety and stress when a larger non-clinical Malaysian sample was examined [41]. Throughout the years, these results were also similar and consistent in numerous samples [40, 42–45].

In terms of construct validity, factor analysis consistently revealed a three-factor structure [41, 45]. As for the concurrent validity, Musa et al. [46] found moderate to strong correlation

with subscales of the Malay version of the Hospital Anxiety and Depressive Scale (HADS) [46]. In this study, the Cronbach's ($\alpha$) value of DASS-21 was 0.90. As for depression, anxiety, and stress scales, they were, respectively, 0.92, 0.89, and 0.90.

## Procedure

**Translation.** After obtaining permission from the original author, the translation process followed the guidelines of Beaton and Bombardier [47]. The instrument underwent forward and backward translation by four multidisciplinary experts. Two translators translated the original version of the instrument (English) into a target language (Malay), and the two translations were then passed to another translators, who then translated the instrument (Malay) into its original language (English). This method is suitable for detecting translation errors and achieving concept equivalence. The instrument was translated into Malay, as it is the national language of Malaysia.

**Content validity.** A report from the translation process was then produced and comparisons were made. The researchers reconciled this translation to select the best wording that would closely match the original DERS-18. For the final verification, two psychology experts and an education expert reviewed the selected items and provided suggestions to strengthen the validity of the instrument.

**Face validity.** Subsequently, three 13-year-old students were provided with the Malay version of the DERS-18 to evaluate the clarity of the items. The participants were instructed to provide a verbal explanation for each item in order to ensure that they accurately assessed the intended construct. The questionnaire was deemed suitable for implementation once the students were able to successfully complete and articulate their understanding of it without encountering any difficulties [48].

**Data collection procedures.** The researchers liaised with the appointed teachers to recruit the students. A study information sheet and parental consent form were distributed to the selected students by the liaison teacher. On the day of the data collection, students who had written parental consent and agreed to take part in the study were requested to complete a set of questionnaires. The questionnaire session lasted for 30 minutes and was conducted in their classroom under the supervision of the liaison teachers. The participants' responses were anonymous and confidential.

**Ethics approval.** The project obtained ethical approval from the Research Ethics Committee (REC) of the Universiti Kebangsaan Malaysia (UKM PPI/111/8/JEP-2021-182). In addition, approvals from the Ministry of Education Malaysia (KPM.600-3/2/3-eras (11775)), the Federal Territory of Kuala Lumpur Education Department (JWPKL.600-9/1/5 JLD.3 (12)), the Selangor Education Department (JPNS.SPO.600-1/1/2 Jld. 14 (56)), the Perlis Education Department (JPNPs.SP. 600-1/1/1 JLD.4 (07)), the Kedah Education Department (JPNK.600-11/1/1 Jld. 2 (49)), the Johor Education Department (JPNJ.PS.600-1/1/2 Jld 11 (59)), the Melaka Education Department (JPNM.SPS.MT6.600-11/1/1 Jld.3 (21)), and the Kelantan Education Department (JPNKN.600-9/1/2 (23)) were also obtained to recruit participants from each school.

## Statistical analysis

All the responses were keyed into IBM-SPSS version 26. Data missing was handled using a listwise deletion, as there was less than 0.02% missing data across the DERS-18 and DASS-21 items [49]. As the data was multivariate non-normal based on the skew and kurtosis results, bootstrapping of maximum likelihood in AMOS version 24 was used.

**Evaluation of floor and ceiling effects.** The floor and ceiling effects indicate the extent to which participants' scores clustered towards the low and high ends of the instrument, respectively. At the item level, the percentage of people who scored 1 (floor effect) and the percentage of participants who scored 5 (ceiling effect) were computed. In relation to the subscale level, the percentage of participants who scored 3 (floor effect) and the percentage of participants who scored 15 (ceiling effect) were calculated. To test the presence of floor and ceiling effect at the total score level, the total scores of DERS-18 with and without "Awareness" were first calculated to find the minimum and maximum scores. Based on the calculations, the percentage of participants who scored 18 (with "Awareness") and 15 (without "Awareness") was calculated to test the presence of the floor effect. Similarly, the percentage of participants who scored 82 (with "Awareness") and 75 (without "Awareness") to test the presence of the ceiling effect. More than 15% of respondents scored minimum or maximum, indicating floor and ceiling effects exist [50].

**Internal consistency reliability.** The internal consistency and reliability of DERS-18 and DASS-21 were tested using Cronbach's α analysis. The values of the Cronbach's α were obtained for the overall DERS-18 and DASS-21, as well as their respective scales.

**Construct validity.** A series of CFA models which are a single factor model, a correlated traits model, a second-order model and a bifactor model confirmatory factor analysis (CFA), were used to confirm the factor structure of the Malay DERS-18. Model fit indices used to evaluate the fit of each model were the relative chi-square ($\chi^2$/df) of less than 5.0 [51], the comparative fit index (CFI), and the Tucker-Lewis index (TLI) of greater than 0.9, and the root mean square error of approximation (RMSEA) of less than 0.06 [51, 52]. Factor loading values above 0.3 indicate the items representing the respective factors [53].

**Criterion validity.** Finally, the concurrent and predictive validity of the Malay version of DERS 18 were tested. A bivariate Pearson's correlation was performed to examine the concurrent validity of the DERS-18 with depression, anxiety and stress in the DASS-21. Following the analysis, a hierarchical regression analysis was run to examine the predictive validity of DERS-18 in predicting depression, anxiety and stress in DASS-21, as in previous studies [4, 19, 23].

## Result

### Content validity and face validity

For the content validity, the three experts concurred the items were appropriate for use with adolescents in Malaysia. Nonetheless, a few Malay words or phrases were modified based on their suggestions to more closely resemble the English terms, as some of the words had been translated literally by the linguists and the intended context was lost. For example, the original translation of "upset" (for items 6 to item 18 of the original instrument) was *susah hati* (worried) or *kecewa* (frust), but *terganggu* (disturbed) has since taken its place. Examples of negative emotions such as sadness (*sedih*), anger (*marah*), and worry (*risau*) were also added to item 6 to item 18, in accordance with the original developers' conceptualization of emotion dysregulation [3, 4]. To enhance intelligibility, sentences were written in the active voice rather than the passive voice. The experts reevaluated and approved the final version of the Malay DERS-18 after the modifications. Regarding the face validity, all the three students provided positive feedback; they had no trouble comprehending the items, were able to respond appropriately, and correctly explained the meaning of each item to field researchers.

### Descriptive results

A total of 701 participants responded to the questionnaire, but twelve participants were excluded due to the missing data on either DERS-18 or DASS-21. Therefore, the final data for

analysis was 689 of 382 girls and 307 boys aged 13 to 14 years old. There were 23.5% of participants from the north region, 21.6% were from the south region, 17.6% were from east region and 37.3% were from the west region of Peninsular Malaysia. The majority of the students were Malays (86.1%), followed by Chinese (7.3%), Indians (3.8%), Bumiputras (2.3%) and immigrants (0.6%). In terms of religion, the majority were Muslims (86.2%). Most of the students were from low to middle income groups and only 4.8% belongs to the high-income group based on Malaysian income classification. Table 1 summarises the participants' demographic characteristics.

## Floor and ceiling effects

Table 2 displays the mean and standard deviation of scores obtained by the participants on the DERS-18 and DASS-21 subscales, along with the floor and ceiling effects for each item, subscale, and total score of DERS-18. At the item level, the vast majority of items exhibited floor effects with percentages ranging from 20.61% to 47.75%. However, items 1, 4 and 6 of the "Awareness" subscale did not exhibit floor effects with percentages ranging from 11.17% to 13.64%. All items exhibited no ceiling effect, with percentages ranging from 5.95% to 13.64%. Item 6, however, demonstrated a ceiling effect of 18.72%.

**Table 1. Demographic characteristics of the participants (N = 689).**

| Demographic characteristics | N | % |
|---|---|---|
| Gender | | |
| Boys | 307 | 44.6 |
| Girls | 382 | 55.4 |
| Age | | |
| 13 years old | 290 | 42.1 |
| 14 years old | 399 | 57.9 |
| Region | | |
| North | 162 | 23.5 |
| South | 149 | 21.6 |
| East | 121 | 17.6 |
| West | 257 | 37.3 |
| Race | | |
| Malay | 593 | 86.1 |
| Chinese | 50 | 7.3 |
| Indian | 26 | 3.8 |
| Bumiputra | 16 | 2.3 |
| Others | 4 | 0.6 |
| Religion | | |
| Islam | 594 | 86.2 |
| Buddha | 43 | 6.2 |
| Hindu | 27 | 3.9 |
| Christian | 9 | 1.3 |
| Others | 16 | 2.3 |
| Monthly family income | | |
| No income / not reported | 18 | 2.6 |
| < RM6000 | 584 | 84.7 |
| RM 6001—RM 10000 | 54 | 7.84 |
| > RM 10001 | 33 | 4.79 |

**Table 2. Means and standard deviations of DASS-21 and DERS-18, as well as skewness, kurtosis, the floor and ceiling effects of DERS-18.**

| | M | SD | Skewness | Kurtosis | Floor Effect (%) | Ceiling Effect (%) |
|---|---|---|---|---|---|---|
| **DASS-21** | | | | | | |
| **Stress** | 10.96 | 6.02 | | | | |
| **Anxiety** | 9.84 | 5.68 | | | | |
| **Depress** | 9.86 | 6.34 | | | | |
| **Awareness** | 9.59 | 2.75 | -0.37 | -0.42 | 2.47 | 1.74 |
| **Item 1** | 3.11 | 1.11 | -0.46 | -0.72 | 11.17 | 5.95 |
| **Item 4** | 3.14 | 1.26 | -0.30 | -1.02 | 13.64 | 12.77 |
| **Item 6** | 3.13 | 1.27 | -0.40 | -0.93 | 11.47 | 18.72 |
| **Clarity** | 7.42 | 3.24 | 0.67 | -0.32 | 10.01 | 4.21 |
| **Item 2** | 2.44 | 1.22 | 0.66 | -0.52 | 24.53 | 8.42 |
| **Item 3** | 2.58 | 1.35 | 0.51 | -0.94 | 25.54 | 13.64 |
| **Item 5** | 2.43 | 1.22 | 0.64 | -0.56 | 24.53 | 7.98 |
| **Non-Acceptance** | 6.83 | 3.09 | 0.86 | 0.03 | 12.92 | 2.47 |
| **Item 7** | 2.38 | 1.25 | 0.74 | -0.45 | 27.86 | 9.43 |
| **Item 13** | 2.15 | 1.22 | 0.98 | -0.01 | 37.16 | 7.40 |
| **Item 14** | 2.33 | 1.23 | 0.75 | -0.40 | 29.61 | 7.98 |
| **Goals** | 7.58 | 3.23 | 0.55 | -0.55 | 9.58 | 2.76 |
| **Item 8** | 2.67 | 1.34 | 0.39 | -1.06 | 22.64 | 13.21 |
| **Item 12** | 2.42 | 1.21 | 0.65 | -0.53 | 24.96 | 7.84 |
| **Item 15** | 2.52 | 1.20 | 0.56 | -0.63 | 20.61 | 7.98 |
| **Impulse** | 6.39 | 3.02 | 0.86 | 0.03 | 19.88 | 1.31 |
| **Item 9** | 2.22 | 1.20 | 0.85 | -0.18 | 33.96 | 6.82 |
| **Item 16** | 2.24 | 1.20 | 0.79 | -0.31 | 32.80 | 6.24 |
| **Item 18** | 1.95 | 1.16 | 1.13 | 0.32 | 47.75 | 4.35 |
| **Strategy** | 6.79 | 2.98 | 0.72 | -0.18 | 13.35 | 1.60 |
| **Item 10** | 2.27 | 1.22 | 0.76 | -0.41 | 32.66 | 7.11 |
| **Item 11** | 2.18 | 1.19 | 0.91 | -0.60 | 35.56 | 6.68 |
| **Item 17** | 2.36 | 1.23 | 0.70 | -0.46 | 28.59 | 8.27 |
| **Total DERS-18** | | | | | | |
| **with Awareness** | 44.60 | 12.18 | 0.66 | 0.09 | 0.15 | 0.15 |
| **without Awareness** | 35.00 | 13.20 | 0.67 | 0.09 | 2.18 | 0.30 |

At the subscale level, all subscales had no floor effects (ranging from 2.47% to 13.35%), except for the impulse subscale (19.88%). There is no ceiling effect for all subscales (ranging from 1.31% to 4.21%). At the total scale level, both total DERS-18 with or without the "Awareness" subscale demonstrated no presence of floor and ceiling effects (<2.18%). All these results indicate good discrimination on the DERS-18 at the total score level, acceptable discrimination at the subscale level, but not at the item level.

## Internal consistencies reliability

Referring to Table 3, the aggregate Cronbach's α value for the Malay DERS-18 was 0.88, with acceptable Cronbach's α values (0.63 to 0.82) for each subscale. All items had a corrected item-total $r$ greater than 0.30 (0.53 to 0.70), with the exception of "Awareness" items (-0.23 to -0.42). The Cronbach's α improved to 0.93 when the "Awareness" items were eliminated. Also enhanced was the corrected item total $r$ (ranging from 0.58 to 0.72). On the basis of these findings, the Malay DERS-18 demonstrated a strong reliability index, particularly

**Table 3. Cronbach's α values and corrected item total r for all the items of the Malay version DERS-18.**

|  | Cronbach's α | Corrected-item Total r | Corrected-item Total r (excluding Awareness) |
|---|---|---|---|
| DERS-18 with Awareness | 0.88 |  |  |
| DERS-18 without Awareness | 0.93 |  |  |
| **Awareness** | 0.63 |  |  |
| Item 1 |  | -0.23 |  |
| Item 4 |  | -0.24 |  |
| Item 6 |  | -0.42 |  |
| **Clarity** | 0.82 |  |  |
| Item 2 |  | 0.62 | 0.63 |
| Item 3 |  | 0.66 | 0.67 |
| Item 5 |  | 0.61 | 0.63 |
| **Non-Acceptance** | 0.79 |  |  |
| Item 7 |  | 0.53 | 0.58 |
| Item 13 |  | 0.66 | 0.69 |
| Item 14 |  | 0.64 | 0.67 |
| **Goals** | 0.82 |  |  |
| Item 8 |  | 0.61 | 0.65 |
| Item 12 |  | 0.69 | 0.71 |
| Item 15 |  | 0.69 | 0.70 |
| **Impulse** | 0.81 |  |  |
| Item 9 |  | 0.69 | 0.71 |
| Item 16 |  | 0.70 | 0.71 |
| Item 18 |  | 0.64 | 0.65 |
| **Strategy** | 0.76 |  |  |
| Item 10 |  | 0.70 | 0.72 |
| Item 11 |  | 0.68 | 0.69 |
| Item 17 |  | 0.58 | 0.61 |

without the "Awareness" items; consequently, these items were eliminated from the subsequent analysis.

### Construct validity

Fig 1 shows the tested models without the "Awareness" subscale and the standardised factor loadings of all items. The unidimensional model did not show an adequate model fit while the five correlated traits model demonstrated an adequate fit. The factor correlations between the five specific factors were moderate to high ($r$ = 0.72 to 0.94). Further, the higher-order model showed an acceptable model fit while the bifactor model exhibited the best fit with smaller RMSEA and $\chi^2/df$ values and larger CFI and TLI values. The fit indices for all the models are summarised in Table 4. Thus, the bifactor model was selected for further analysis.

In the bifactor model, all the items demonstrated higher factor loadings on the general factor than on their specific factor (see Fig 1 and Table 5). All the items loaded more than 0.30 on the general factor ($\lambda$ = 0.66 to 0.77). For the specific factor-target loadings, the "Clarity", "Goals", and "Non-Acceptance" subscales exhibited moderate factor loadings from 0.31 to 0.67, except for item 14 ($\lambda$ = 0.26). Otherwise, all "Impulse" items ($\lambda$ = 0.12 to 0.17) and "Strategies" items, which were item 10 ($\lambda$ = 0.19) and item 17 ($\lambda$ = -0.13), had low factor loadings. This result indicated "Impulse" and "Strategy" have low specificity after controlling the

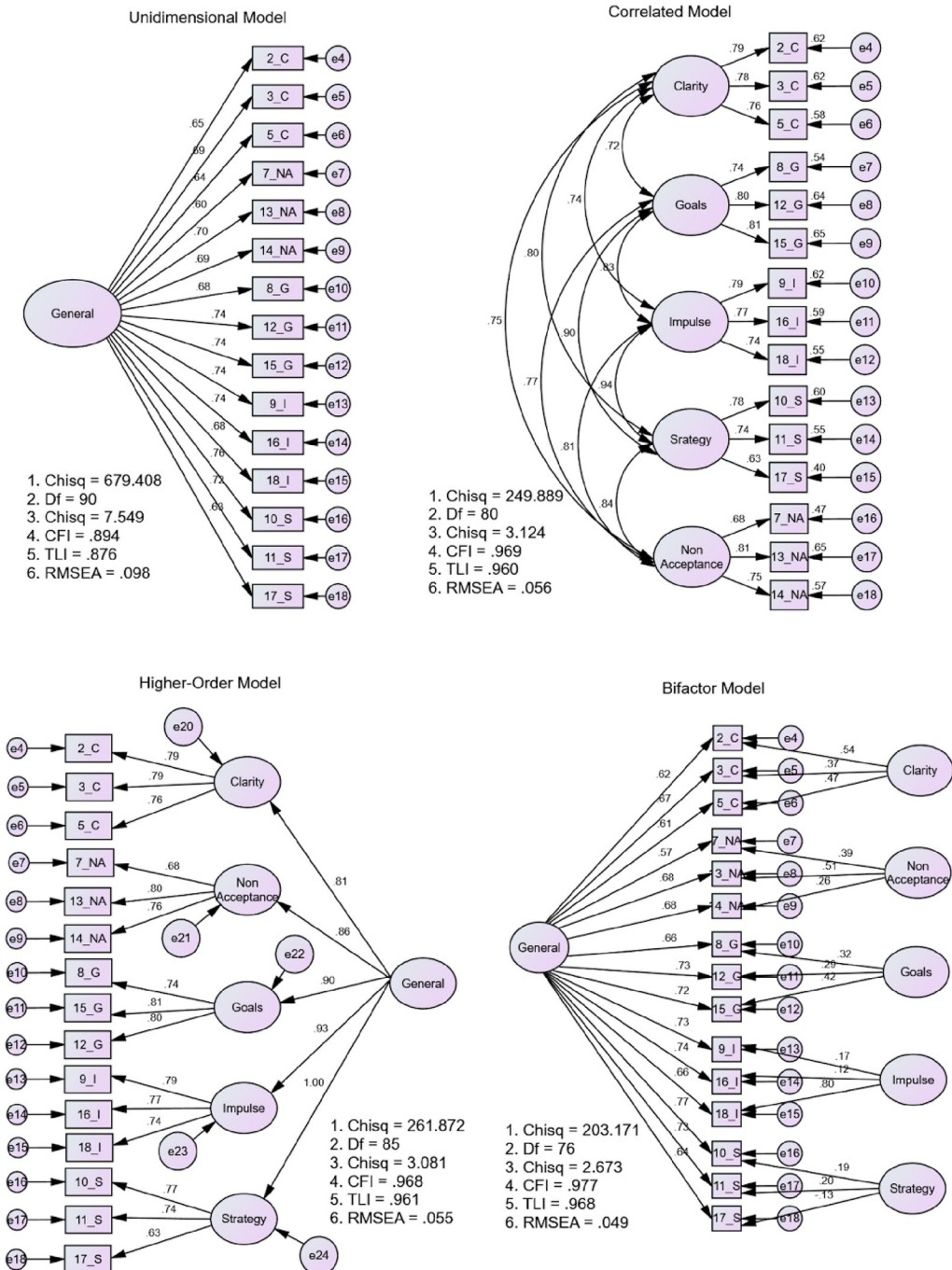

**Fig 1. The tested CFA models to confirm the dimensionality of the Malay DERS-18.** The paths shown were standardised factor loadings.

**Table 4.  Model fit indices of DERS-18 without "Awareness".**

| Model | $\chi^2$ (*df*) | $\chi^2$/*df* | RMSEA | CFI | TLI | *Δ*RMSEA | *Δ*CFI |
|---|---|---|---|---|---|---|---|
| Unidimensional | 679.408 (90) | 7.549 | 0.098 | 0.894 | 0.876 | | |
| Correlated traits | 249.889 (80) | 3.124 | 0.056 | 0.969 | 0.960 | -0.042 | 0.075 |
| Higher-order | 261.872 (85) | 3.081 | 0.055 | 0.968 | 0.961 | -0.001 | 0.001 |
| Bifactor | 203.171 (76) | 2.673 | 0.049 | 0.977 | 0.968 | -0.006 | 0.009 |

variance explained by the general factor. For this reason, the "Clarity", "Goals", and "Non-Acceptance" subscales were retained for further analysis.

## Criterion validity

Table 6 presents the observed correlations between the DERS-18 subscales, the DERS-18 total score, and the DASS-21 subscales. All subscales moderately and significantly correlated with each other in a positive direction. DERS-18 without "Awareness" was significantly and positively associated with stress ($r = 0.64$, $p < 0.01$), anxiety ($r = 0.62$, $p < 0.01$), and depression ($r = 0.64$, $p < 0.01$). All the subscales were significantly correlated with "Awareness" in a negative direction. These results indicated the DERS-18 without "Awareness" portrayed concurrent validity.

Subsequently, the hierarchical regression analysis was conducted to examine the specific attribution "Clarity", "Non-Acceptance", and "Goals" in predicting psychological symptoms in DASS-21 [4, 22, 31]. Stress, anxiety and depression scores were set as dependent variables. Age and gender were entered at Step 1, and following the total of DERS-18 scores (general factors) entered at Step 2, the model explained 42.9% to 45.1% of the variance in stress, anxiety, and depression. At Step 3, "Clarity", "Non-acceptance", and "Goals" (specific factors) were entered, the model explained an additional 0.7% to 1.7% of the total variance of stress, anxiety, and depression. These findings showed the DERS-18 and its specific factors ("Clarity", "Non-Acceptance", and "Goals") demonstrated predictive validity. The summary of the results is shown in Table 7.

## Discussion

The current study investigated the psychometric properties of the Malay version of the DERS-18. The DERS-18 was translated into Malay through the implementation of both forward and backward translations techniques. The Malay DERS-18 underwent content and face validity assessments by both experts in the field and adolescents. Subsequently, the DERS-18 assessment tool was implemented among students between the ages of 13 and 14 years who enrolled in schools located in Peninsular Malaysia. In general, the Malay DERS-18 scale exhibited sufficient sensitivity to measure emotion dysregulation. Moreover, the DERS-18 excluding the "Awareness" subscale, portrayed excellent reliability in terms of internal consistency, and demonstrated sound validity in both construct and criterion.

Firstly, this study found a floor effect at the item level (except for the "Awareness" items), but not at the subscale level (except for "Impulse") and at the total level. This effect may be due to the fact that our sample consisted of non-clinical adolescents who did not experience high levels of emotion dysregulation. As a result, many of them scored the minimum score on the DERS-18 items. However, the DERS-18 maybe more sensitive to specific aspects of emotion dysregulation, except for impulse control. Adolescents in this study may generally experience less difficulty with impulse control compared to other aspects of emotion dysregulation. Since

**Table 5. The Malay translation of DERS-18 and the standardised factor loading of DERS-18 without "Awareness" across the CFA models.**

| DERS-18 Item/Model | Standardised factor loading | | | | | |
| --- | --- | --- | --- | --- | --- | --- |
| | Single Factor | Correlated | Second Order | | Bifactor | |
| | | | First | Second | General | Specific |
| **Awareness** | | | | | | |
| 1. I pay attention to how I feel. *Saya memberi perhatian terhadap apa yang saya rasa.* | - | - | - | - | - | - |
| 4. I am attentive to my feelings. *Saya peka terhadap perasaan saya.* | - | - | - | - | - | - |
| 6. When I'm upset, I acknowledge my emotions. *Saya mengakui perasaan saya apabila saya rasa terganggu (sedih, marah, risau dan sebagainya).* | - | - | - | - | - | - |
| **Clarity** | | | 0.81 | | | |
| 2. I have no idea how I am feeling. *Saya tidak tahu apakah sebenarnya yang saya rasa.* | 0.65 | 0.79 | | 0.79 | 0.62 | 0.54 |
| 3. I have difficulty making sense out of my feelings. *Saya mengalami kesukaran untuk memahami isi hati saya sendiri.* | 0.69 | 0.78 | | 0.79 | 0.67 | 0.37 |
| 5. I am confused about how I feel. *Saya keliru dengan apa yang saya rasa.* | 0.64 | 0.76 | | 0.76 | 0.61 | 0.47 |
| **Non-Acceptance** | | | 0.86 | | | |
| 7. When I'm upset, I become embarrassed for feeling that way. *Saya berasa malu apabila saya rasa terganggu (sedih, marah, risau dan sebagainya).* | 0.60 | 0.69 | | 0.68 | 0.57 | 0.39 |
| 13. When I'm upset, I feel ashamed with myself for feeling that way. *Saya berasa malu dengan diri sendiri apabila saya rasa terganggu (sedih, marah, risau dan sebagainya).* | 0.70 | 0.81 | | 0.80 | 0.68 | 0.51 |
| 14. When I'm upset, I feel guilty for feeling that way. *Saya berasa serba salah apabila saya rasa terganggu (sedih, marah, risau dan sebagainya).* | 0.69 | 0.75 | | 0.76 | 0.68 | 0.26 |
| **Goals** | | | 0.90 | | | |
| 8. When I'm upset, I have difficulty getting work done. *Saya mengalami kesukaran menyiapkan tugasan apabila saya rasa terganggu (sedih, marah, risau dan sebagainya).* | 0.68 | 0.74 | | 0.74 | 0.66 | 0.32 |
| 12. When I'm upset, I have difficulty focusing on other things. *Saya mengalami kesukaran untuk memberikan perhatian kepada perkara-perkara yang lain apabila saya rasa terganggu (sedih, marah, risau dan sebagainya).* | 0.74 | 0.80 | | 0.81 | 0.73 | 0.29 |
| 15. When I'm upset, I have difficulty concentrating. *Saya mengalami kesukaran untuk menumpukan perhatian apabila saya rasa terganggu (sedih, marah, risau dan sebagainya).* | 0.74 | 0.81 | | 0.80 | 0.72 | 0.42 |
| **Impulse** | | | 0.93 | | | |
| 9. When I'm upset, I become out of control. *Saya menjadi tidak terkawal apabila saya rasa terganggu (sedih, marah, risau dan sebagainya).* | 0.74 | 0.79 | | 0.79 | 0.73 | 0.17 |
| 16. When I'm upset, I have difficulty controlling my behaviours. *Saya mengalami kesukaran untuk menjaga tingkah laku saya apabila rasa terganggu (sedih, marah, risau dan sebagainya).* | 0.74 | 0.77 | | 0.77 | 0.74 | 0.12 |
| 18. When I'm upset, I lose control over my behaviours. *Saya hilang kawalan terhadap tingkah laku saya apabila rasa terganggu (sedih, marah, risau dan sebagainya).* | 0.68 | 0.74 | | 0.74 | 0.66 | 0.80 |
| **Strategy** | | | 1.00 | | | |
| 10. When I'm upset, I believe that I will remain that way for a long time. *Saya percaya saya akan berasa terganggu untuk masa yang lama apabila saya rasa terganggu (sedih, marah, risau dan sebagainya).* | 0.76 | 0.78 | | 0.77 | 0.77 | 0.19 |
| 11. When I'm upset, I believe that I'll end up feeling very depressed. *Saya percaya bahawa diri saya akan berasa sangat murung apabila saya rasa terganggu (sedih, marah, risau dan sebagainya).* | 0.72 | 0.74 | | 0.74 | 0.73 | 0.20 |
| 17. When I'm upset, I believe that wallowing in it is all I can do. *Saya percaya yang saya hanya mampu melayan perasaan saya apabila rasa terganggu (sedih, marah, risau dan sebagainya).* | 0.63 | 0.63 | | 0.63 | 0.64 | -0.13 |

**Table 6. Correlations among subscales in DERS-18 and DASS-21.**

|  | 1 | 2 | 3 | 4 | 5 | 6 | 7 | 8 | 9 | 10 |
|---|---|---|---|---|---|---|---|---|---|---|
| **1. Awareness** |  |  |  |  |  |  |  |  |  |  |
| **2. Clarity** | -0.35 |  |  |  |  |  |  |  |  |  |
| **3. Non-Acceptance** | -0.45 | 0.61 |  |  |  |  |  |  |  |  |
| **4. Goals** | -0.39 | 0.59 | 0.62 |  |  |  |  |  |  |  |
| **5. Impulse** | -0.33 | 0.60 | 0.64 | 0.68 |  |  |  |  |  |  |
| **6. Strategy** | -0.43 | 0.65 | 0.66 | 0.71 | 0.74 |  |  |  |  |  |
| **7. DERS-18 (with Awareness)** | -0.28 | 0.81 | 0.80 | 0.84 | 0.86 | 0.86 |  |  |  |  |
| **8. DERS-18 (without Awareness)** | -0.46 | 0.82 | 0.83 | 0.85 | 0.86 | 0.88 | 0.98 |  |  |  |
| **9. Stress** | -0.39 | 0.60 | 0.56 | 0.57 | 0.55 | 0.57 | 0.64 | 0.67 |  |  |
| **10. Anxiety** | -0.37 | 0.58 | 0.55 | 0.55 | 0.53 | 0.57 | 0.62 | 0.65 | 0.88 |  |
| **11. Depress** | -0.34 | 0.62 | 0.54 | 0.56 | 0.51 | 0.59 | 0.64 | 0.67 | 0.88 | 0.86 |

All the correlations were significant at the 0.01 level

there were no floor effects at the total level and no ceiling effects at any level, this suggests that the DERS-18 as a whole is still able to distinguish between individuals with different levels of emotion dysregulation.

The overall reliability value of the Malay DERS-18 was comparable to that of previous DERS-18 validation studies [4, 18, 19, 22–24]. Internal consistency and reliability of the total scale and subscales were moderate to high, according to this study's findings. The reliability value improved further after the "Awareness" items were eliminated. Additionally, the "Awareness" subscale was negatively correlated with the other DERS-18 and DASS-21 subscales. Theoretically, the "Awareness" subscale should have a positive correlation with other DERS-18 subscales, indicating emotion dysregulation.

The usefulness of "Awareness" in conceptualizing and measuring deficits in emotion regulation has been debated in previous research [20, 22, 23]. Osborne et al. [17] noted that "Awareness" is the only subscale containing reverse-coded items, which could compromise the psychometric properties of DERS-18. This was not necessarily the case, however, when Bardeen et al. [54] rephrased the "Awareness" items using positive language. They discovered that the "Awareness" items loaded alongside the "Clarity" items. Possible explanations for the Malay DERS-18 were that probably the adolescents had trouble responding appropriately to the negatively phrased items [55] or that their responses did not accurately reflect who they are [56]. In addition, linguistic translation of reverse-scored items may convey distinct meanings and typically result in subpar psychometric properties [56, 57].

**Table 7. The total variance of stress, anxiety, and depression explained by the total scale and subscale scores of the Malay version of the DERS-18.**

| | $R^2$ | | $\Delta R^2$ |
|---|---|---|---|
| **Criterion** | **General Factor** | **General and Specific Factor** | |
| **Stress** | 0.45 | 0.46 | 0.01 |
| **Anxiety** | 0.43 | 0.44 | 0.01 |
| **Depression** | 0.44 | 0.46 | 0.02 |

All the $\Delta R^2$ were statistically significant at $p < 0.001$

Therefore, it is possible to administer the Malay DERS-18 without the "Awareness" subscale to Malaysian adolescents. As an alternative, the items of the "Awareness" subscale must be reworded in a positive manner [13]. As mentioned by a few other researchers of specific populations and countries, the significance of the "Awareness" subscale in the overall emotional dysregulation construct is also dubious [13, 14, 17, 26, 27].

In terms of construct validity, findings from CFA revealed that the bifactor model for the Malay version of the DERS-18 without "Awareness" was superior to other models. The bifactor model allows researchers to examine the unidimensionality and multidimensionality of an instrument simultaneously [58]. This model can facilitate the theoretical clarity of emotion dysregulation and also evaluate the soundness of each DERS-18 subscale in explaining traits of emotion dysregulation [59]. Consistent with previous studies that found the bifactor model fit to their data [21–23, 27], this study supported the use of a total score in measuring emotion dysregulation and also measured the single traits of emotion dysregulation simultaneously [58].

In terms of criterion validity, this study's findings demonstrated concurrent and predictive validity. For concurrent validity, both total scales and subscales of DERS-18 (not "Awareness") showed a positive association with the psychological symptoms as expected [7, 8]. Nevertheless, it is noteworthy that this study's findings for predictive validity showed that the "Clarity", "Goals" and "Non-Acceptance" subscales explained the variance in predicting the psychological symptoms, which was consistent with the findings by Zhao et al. [21]. Thus, the use of "Clarity", "Goals" and "Non-Acceptance" subscales together with the total score of the DERS-18 are recommended among Malaysian adolescents.

This study has several limitations that must be addressed. First, the present study began by recruiting a non-clinical sample of adolescents. It is possible to increase the generalizability of findings by recruiting individuals who are more prone to emotion regulation difficulties, such as clinical patients (e.g., those diagnosed with bipolar disorder). Second, future studies may replicate this result by recruiting samples across adolescent stages (e.g., early, middle and late adolescents) who may respond appropriately to the "Awareness" items and also be able to discriminate between the items. Third, the sample may not fully be represented of all Malaysian adolescents due to the fact that most of the respondents were Malays. In reality, Malaysia is a diverse nation, comprised of East and Peninsular Malaysia, each of which has distinct ethnic groups. This study was conducted on national school students, the majority of whom are of Malay decent. Consider including more participants of other ethnicities, such as Chinese and Indians, in future research.

Lastly, this study was unable to conduct a test-retest reliability analysis due to the COVID-19 pandemic. The COVID -19 pandemic was a highly stressful and unpredictable event, and it is likely that the emotional state of participants would have fluctuated. Therefore, their response to the DERS-18 items might not be consistent over time. We believed that the test-retest results would not be reliable in which the scores of the first and second time might not be correlated and thus, may falsely indicate the instability of the DERS-18. Additionally, the restricted Movement Control Order was imposed by the Malaysian Government made it difficult to ensure that the same participants would be available for the test-retest session. It is recommended that future studies conduct test-retest reliability analysis to provide further evidence of the validity of the Malay version of DERS-18.

In conclusion, the current study has provided insights regarding the psychometric attributes of the Malay version of the DERS-18. The findings illustrate that the Malay DERS-18, when excluding the "Awareness" subscale, is a reliable and sound tool for evaluating emotion regulation deficiencies in Malaysian adolescents. This research endeavour serves as a foundational basis for Malaysian studies to investigate challenges in regulating emotions through the

use of a validated instrument. This will facilitate the understanding of the nature of emotion regulation among Malaysians.

## Supporting information

**S1 Dataset.**
(SAV)

## Author Contributions

**Conceptualization:** Nur Afrina Rosharudin, Tuti Iryani Mohd Daud, Suzana Mohd Hoesni, Siti Rashidah Yusoff.

**Data curation:** Nur Afrina Rosharudin, Siti Rashidah Yusoff, Mohamad Omar Ihsan Razman.

**Formal analysis:** Nur Afrina Rosharudin.

**Funding acquisition:** Manisah Mohd Ali.

**Investigation:** Nur Afrina Rosharudin, Noor Azimah Muhammad, Tuti Iryani Mohd Daud, Suzana Mohd Hoesni, Siti Rashidah Yusoff, Mohamad Omar Ihsan Razman, Manisah Mohd Ali, Khairul Farhah Khairuddin, Dharatun Nissa Puad Mohd Kari.

**Methodology:** Nur Afrina Rosharudin, Noor Azimah Muhammad, Tuti Iryani Mohd Daud, Manisah Mohd Ali.

**Project administration:** Noor Azimah Muhammad, Suzana Mohd Hoesni, Manisah Mohd Ali, Khairul Farhah Khairuddin, Dharatun Nissa Puad Mohd Kari.

**Resources:** Noor Azimah Muhammad, Tuti Iryani Mohd Daud, Suzana Mohd Hoesni, Manisah Mohd Ali.

**Supervision:** Suzana Mohd Hoesni, Manisah Mohd Ali.

**Validation:** Noor Azimah Muhammad, Tuti Iryani Mohd Daud.

**Writing – original draft:** Nur Afrina Rosharudin, Siti Rashidah Yusoff, Mohamad Omar Ihsan Razman.

**Writing – review & editing:** Nur Afrina Rosharudin, Noor Azimah Muhammad, Tuti Iryani Mohd Daud, Siti Rashidah Yusoff, Mohamad Omar Ihsan Razman.

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
