## [Decision Letter · Decision Letter 0]

30 Mar 2023

PONE-D-22-33847Utilizing a Bifactor Model in Validating the Malay version of the Difficulties in Emotion Regulation Scale-18 in Malaysian AdolescentsPLOS ONE

Dear Dr. Muhammad,

Thank you for submitting your manuscript to PLOS ONE. After careful consideration, we feel that it has merit but does not fully meet PLOS ONE’s publication criteria as it currently stands. Therefore, we invite you to submit a revised version of the manuscript that addresses the points raised during the review process.

 Please submit your revised manuscript by May 14 2023 11:59PM. If you will need more time than this to complete your revisions, please reply to this message or contact the journal office at plosone@plos.org. Please include the following items when submitting your revised manuscript:A rebuttal letter that responds to each point raised by the academic editor and reviewer(s). You should upload this letter as a separate file labeled 'Response to Reviewers'.A marked-up copy of your manuscript that highlights changes made to the original version. You should upload this as a separate file labeled 'Revised Manuscript with Track Changes'.An unmarked version of your revised paper without tracked changes. You should upload this as a separate file labeled 'Manuscript'.

We look forward to receiving your revised manuscript.

Kind regards,

Qi Yuan

Academic Editor

PLOS ONE

Journal Requirements:

"We would like to express our deepest gratitude to the Ministry of Higher education (MoHE) Malaysia for enabling this research to be implemented and supported financially (Transdisciplinary Research Grant Scheme, TRGS/1/2020/UKM/01/4/1). "

"Manisah Mohd Ali received a fund from Ministry of Higher education (MoHE) Malaysia (Project code: TRGS/1/2020/UKM/01/4/1). The funders had no role in study design, data collection and analysis, decision to publish, or preparation of the manuscript"

Additional Editor Comments:

The article itself is interesting, but I also agree with the reviewer that it seems more like a traditional CFA paper. Please make necessary changes and also address the comments from the reviewers.

Reviewers' comments:

Reviewer's Responses to Questions

**Comments to the Author**

1. Is the manuscript technically sound, and do the data support the conclusions?

Reviewer #1: Yes

Reviewer #2: Partly

2. Has the statistical analysis been performed appropriately and rigorously? 

Reviewer #1: Yes

Reviewer #2: No

3. Have the authors made all data underlying the findings in their manuscript fully available?

Reviewer #1: Yes

Reviewer #2: No

4. Is the manuscript presented in an intelligible fashion and written in standard English?

Reviewer #1: No

Reviewer #2: No

5. Review Comments to the Author

Reviewer #1: "Emotional regulation" refers to the ability to be self-aware, process emotions, and respond appropriately. The article advances our understanding of current issues with emotion regulation, particularly in adolescents. With a few simple adjustments, the paper could be made better.

1. There are a few grammatical errors, incorrect spacing, and a full stop throughout the paper.

2. There was inconsistency regarding the study sample size. For instance, 701, 700, and 689.

3. The paragraph on obtaining ethical approval should be under the heading "Ethical approval."

4. The authors stated that "A random stratified sampling was used to select the schools from each region." The authors should provide some information about this process.

Reviewer #2: Dear author

The following points are suggested

I am not agree with the title, you should structure it conventionally according to studies done worldwide about psychometric evaluation of instruments, why bifactors; this is a result from CFA, Please revise title by following conventional titles in this area!

Abstract

This section in all subsection should be revised and completed according to the following suggestions, for example methods subsection is not complete and you mentioned partly some aspects of psychometric analysis done on needed to be done.

I am not agree with predictive validity, it is criterion validity (divergent/convergent validity)

How about the test retest reliability results

The presented data about criterion validity is not correct, why r-squared not r and how about p-value for r.

Conclusion does not present any thing about construct validity.

not suitable keyword :Malay validation, validity or psychometric analysis, you can include Malesia as a keyword separately, use construct validity instead of confirmatory factor analysis

Introduction is very long, should be effectively shorten with more focuses on conducted studies worldwide about validity and reliability of DERS-18

Methods and results

Sampling methods to my view is not correct, if you used stratified sampling what were the strata? generally the participants section should be revised extensively by explaining the more and correct details about the sampling process.

instruments

Total scales wrong term?!

DERS-18 should be explained to the readers with more details from developers.

More details about psychometric properties of DASS-21 from Malesia are needed.

you explained very brief information about psychometric evaluation in statistical analysis subsection, you should subdivide this section into subsections with heading such as "construct validity" , internal consistency and item analysis" , "criterion validity" and explain the process for each of above procedures with more details,

Surprisingly a very important step i.e. test retest reliability has been missed,

Ceiling and flor effect evaluation have been missed,

How about the process of face and content validity evaluation and their results

For all above points i strongly recommend authors to see published papers on psychometric evaluation in good refereed journals for obtaining guides on how to write and complete the above mentioned shortfalls

I is expected to present a table explaining the study participants characteristics including basic such as age, gender, education,... and the mean scores of instruments subscales you used in this study,

Table 1 does not present informative data in the framework of psychometric analysis, particularly correlations.

The results about reliability analysis has major defects, the most important you did not follow the union approach for treating the awareness domain , you should follow the original version for scoring and then follow the reliability analysis, please revise and reconsider this very important point.

Skewness Kurtosis are not informative data in this regard.

Please present the results of test retest analysis.

Please (after reconsidering awareness domain) present CFA results in a suitable figure as is very common in this area of subject, the correct presentation as table is confusing.

Please follow my point regarding the criterion validity in abstract, and see published papers for getting guides on how to present and interpret the results of criterion validity.

Please present the results of Ceiling and flor effect evaluations.

6. PLOS authors have the option to publish the peer review history of their article (what does this mean?). If published, this will include your full peer review and any attached files.

Reviewer #1: No

Reviewer #2: No

---

## [Author Response · Author response to Decision Letter 0]

23 May 2023

A) Addressing editor 1's comments 

1. In the Methods section please include the informed consent statement to reflect whether "written or verbal" informed consent was obtained from all participants for inclusion in the study.

- We have added the "written parental consent" under the Methods section

2. However, please note that it is not acceptable for an author to be the sole named individual responsible for ensuring data access.... If legal or ethical restrictions apply, please provide all necessary instructions and non-author contact information (preferably email) for a data access committee, ethics committee, or other institutional body that other researchers would require to request access to your data.

- we propose the following: (1) The minimal anonymized dataset will remain publicly available in the Supporting Information, (2) The complete dataset, including potentially sensitive information, will be made accessible upon request, and (3) To facilitate data access, we had a data access committee consisting of the corresponding author, Noor Azimah Muhammad (drazimah@ppukm.edu.my), the main author, Nur Afrina Rosharudin (p111600@siswa.ukm.edu.my), one of the authors, Suzana Mohd Hoesni (smh@edu.ukm.my) and a designated group representative, TRGS Let’s Get R.E.A.L (trgsletsgetreal@gmail.com).

B) Addressing editor 2's comments 

1. Please ensure that your manuscript meets PLOS ONE's style requirements, including those for file naming. The PLOS ONE style templates can be found at..

- We have followed the requirements and made amendments

2. Please remove any funding-related text from the manuscript and let us know how you would like to update your Funding Statement. Currently, your Funding Statement reads as follows...

- We have removed the text from our manuscript and agreed with the suggested statement

3) If there are ethical or legal restrictions on sharing a de-identified data set, please explain them in detail... Please also provide contact information for a data access committee, ethics committee, or other institutional body to which data requests may be sent

- the data is not identifying participants per se but may contain potentially sensitive information such as the name of school which was involved in this study

- the complete set of the data can be requested from the following authors: Noor Azimah Muhammad (drazimah@ppukm.edu.my), Suzana Mohd Hoesni (smh@ukm.edu.my), and Manisah Mohd Ali (drmanisah.ma@gmail.com)

4. If there are no restrictions, please upload the minimal anonymized data set necessary to replicate your study findings as either Supporting Information files or to a stable, public repository and provide us with the relevant URLs, DOIs, or accession numbers

- we have uploaded the minimal anonymized data set as Supporting information

5. If you wish to make changes to your Data Availability statement, please describe these changes in your cover letter and we will update your Data Availability statement to reflect the information you provide

- No change

6. please upload your figure files to the Preflight Analysis and Conversion Engine (PACE) digital diagnostic tool...

- we have used the PACE to download the figure file 

C) Addressing Reviewer 1's Comments

1. There are a few grammatical errors, incorrect spacing, and a full stop throughout the paper.

- We have sent the manuscript for proofreading and editing

2. There was inconsistency regarding the study sample size. For instance, 701, 700, and 689

- The size sample was different at the data collection (701 participants) and at the final analysis (689 participants) because of missing data. We added a clarification at line 197, 217 to 219, and 387 to 390.

3. The paragraph on obtaining ethical approval should be under the heading "Ethical approval."

- Changes have been made at line, see line 297 to 300. From line 308 to 317 is the paragraph on obtaining ethical approval

4. The authors stated that "A random stratified sampling was used to select the schools from each region." The authors should provide some information about this process. 

- We added a clarification from line 203 to 212.

D) Addressing Reviewer 2's Comments

5. Abstract: This section in all subsection should be revised and completed according to the following suggestions, for example methods subsection is not complete and you mentioned partly some aspects of psychometric analysis done on needed to be done.

- We have added analysis procedure See line 33 to 37

6. Abstract: I am not agree with predictive validity, it is criterion validity (divergent/convergent validity)

- The word predictive have been replaced with criterion (concurrent and predictive). See line 50

7. Abstract: How about the test retest reliability results?

- This study was conducted during the Covid-19 pandemic, the adolescents emotional state may not be stable would be affected by the unprecedented event over a period of time. We did not perform the test-retest study (that is mainly to look into the replicability of the responses) as we felt it is not applicable for the current study. Hence, the result is not available. In addition, as the data collection was conducted during restricted Movement Control Order of the Covid-19 pandemic, it was difficult to ensure to catch the same students for the re-test session as some students may be absent from the (online or physical) classes because of health reasons, school truancy or internet issues. Their academic activities were very much affected and it would be difficult for us to arrange meetings of more than once as the priority was more on more on completing their syllabus.

8. Abstract: The presented data about criterion validity is not correct

- We have made the correction. We subdivide the criterion validity into concurrent validity and predictive validity. See line 44 to 48

9. Abstract - why r-squared not r and

- The reason is that previous studies had examined the predictive validity of the DERS-18 using regression analysis. Thus, we aimed to replicate the findings

10. Abstract: how about p-value for r.

- We added the results of r and p-value. See line 44 to 46

11. Abstract: Conclusion does not present any thing about construct validity.

- Have added in the conclusion. See line 50

12. not suitable keyword :Malay validation, validity or psychometric analysis, you can include Malesia as a keyword separately, use construct validity instead of confirmatory factor analysis

- Changes have been made. See line 53 to 54

13. Intro: Introduction is very long, should be effectively shorten with more focuses on conducted studies worldwide about validity and reliability of DERS-18

- We have shortened the introduction, while retaining important information such as the dimensionality and factor structure of the DERS-18. We believe these details are crucial to discuss later in the discussion section. See line 58 to 194

14. Methods: Sampling methods to my view is not correct, if you used stratified sampling what were the strata? generally the participants section should be revised extensively by explaining the more and correct details about the sampling process

- Have added a clarification from line 203 to 212

15. Methods: instruments . Total scales wrong term?

- Have replaced it with a correct term, see line 243

16. Methods: DERS-18 should be explained to the readers with more details from developers.

- Have added an explanation. From line 226 to 245

17. Methods: More details about psychometric properties of DASS-21 from Malaysia are needed

- Have added an explanation. From 246 to 264

18. Methods: you explained very brief information about psychometric evaluation in statistical analysis subsection, you should subdivide this section

into subsections with heading such as "construct validity" , internal consistency and item analysis" , "criterion validity" and explain the process for each of above procedures with more detail

- Have added the subsections. From 319 to 363

19. Methods: Surprisingly a very important step i.e. test-retest reliability has been missed

- The study was conducted during the Covid-19 pandemic, which may have affected the emotional state of adolescents over time. A test-retest study was not conducted as it was deemed impractical. The data collection was also affected by the pandemic, as some students may have been absent from classes due to various reasons, making it difficult to arrange for a re-test session. Additionally, academic activities were disrupted due to the pandemic, making it harder to arrange multiple meetings. Thus, the data was not available.

20. Methods: Ceiling and floor effect evaluation have been missed

- Have added the evaluation. From line 325 to 339

21. Methods/Results: How about the process of face and content validity evaluation and their results

- Have added explanations on the process of evaluating the face and content validity. See line 277 to 294. The results of these validity are shown in line 366 to 384

22. Results: For all above points i strongly recommend authors to see published papers on psychometric evaluation in good refereed journals for obtaining guides on how to write and complete the above mentioned shortfalls 

- Have followed the published papers on psychometric evaluation especially papers from PLOS One 

23. Results: I is expected to present a table explaining the study participants characteristics including basic such as age, gender, education,...

- Have added the description of participant characteristics. See line 387 to 396. We also added a table (Table 1) at line 398

24. Results: and the mean scores of instruments subscales you used in this study,

- We have added a table (Table 2) at line 424

25. Results: Table 1 does not present informative data in the framework of psychometric analysis, particularly correlations

- The table containing correlation has been excluded. See line 401 to 405

26. Results: The results about reliability analysis has major defects, the most important you did not follow the union approach for treating the awareness domain , you should follow the original version for scoring and then follow the reliability analysis, please revise and reconsider this very important point

- We have followed the suggestion. See line 427 until 444

27. Results: Skewness Kurtosis are not informative data in this regard.

 - Have excluded the information. See See Table 3 at line 446 

28. Results: Please present the results of test retest analysis

- Due to the pandemic, it was deemed impractical to conduct a test-retest study and data collection was affected by student absences. Academic activities were also disrupted, making it challenging to arrange multiple meetings.

29. Results: Please (after reconsidering awareness domain) present CFA results in a suitable figure as is very common in this area of subject, the correct presentation as table is confusing

- We have uploaded a figure (Figure 1) as an individual file that is separate from the manuscript. The figure will be inserted at line 462

30. Please follow my point regarding the criterion validity in abstract, and see published papers for getting guides on how to present and

interpret the results of criterion validity

- We have presented the results of criterion validity. For the concurrent validity, see line 482 to 491. For the predictive validity, see line 493 to 502

31. Please present the results of Ceiling and flor effect evaluations.

- Presented at line 407 to 422

---

## [Decision Letter · Decision Letter 1]

21 Jul 2023

Psychometric Properties of the Malay version of the Difficulties in Emotion Regulation Scale-18 in Malaysian Adolescents

PONE-D-22-33847R1

Dear Dr. Muhammad,

We’re pleased to inform you that your manuscript has been judged scientifically suitable for publication and will be formally accepted for publication once it meets all outstanding technical requirements.

Kind regards,

Qi Yuan

Academic Editor

PLOS ONE

Additional Editor Comments (optional):

Dear authors,

Thank you very much for your revision. Please remember to address the minor comments from review 2 during proof.

Sincerely,

Qi

Reviewers' comments:

Reviewer's Responses to Questions

**Comments to the Author**

1. If the authors have adequately addressed your comments raised in a previous round of review and you feel that this manuscript is now acceptable for publication, you may indicate that here to bypass the “Comments to the Author” section, enter your conflict of interest statement in the “Confidential to Editor” section, and submit your "Accept" recommendation.

Reviewer #1: All comments have been addressed

Reviewer #2: All comments have been addressed

2. Is the manuscript technically sound, and do the data support the conclusions?

Reviewer #1: Yes

Reviewer #2: Yes

3. Has the statistical analysis been performed appropriately and rigorously? 

Reviewer #1: Yes

Reviewer #2: Yes

4. Have the authors made all data underlying the findings in their manuscript fully available?

Reviewer #1: Yes

Reviewer #2: Yes

5. Is the manuscript presented in an intelligible fashion and written in standard English?

Reviewer #1: Yes

Reviewer #2: Yes

6. Review Comments to the Author

Reviewer #1: I am satisfied with the author's correction, and I believe the manuscript can be accepted for publication.

Reviewer #2: Please mention undoing of test-retest reliability as a study limitation. Also, please mention undoing of content and face validity evaluation too.

7. PLOS authors have the option to publish the peer review history of their article (what does this mean?). If published, this will include your full peer review and any attached files.

Reviewer #1: No

Reviewer #2: **Yes: **Dr.Awat Feizi

---

## [Editor Report · Acceptance letter]

18 Aug 2023

PONE-D-22-33847R1 

Psychometric Properties of the Malay version of the Difficulties in Emotion Regulation Scale-18 in Malaysian Adolescents 

Dear Dr. Muhammad:

I'm pleased to inform you that your manuscript has been deemed suitable for publication in PLOS ONE. Congratulations! Your manuscript is now with our production department. 

Kind regards, 

on behalf of

Dr. Qi Yuan 

Academic Editor

PLOS ONE